# Exploring Progesterone Deficiency in First-Trimester Miscarriage and the Impact of Hormone Therapy on Foetal Development: A Scoping Review

**DOI:** 10.3390/children11040422

**Published:** 2024-04-02

**Authors:** Munkhtuya Bataa, Erini Abdelmessih, Fahad Hanna

**Affiliations:** 1Public Health Program, Department of Health and Education, Torrens University Australia, Melbourne 3000, Australia; munkhtuya.bataa@health.torrens.edu.au; 2School of Health Science, The University of Notre Dame, Sydney 2007, Australia; erini.abdelmessih@utas.edu.au

**Keywords:** pregnancy, first trimester, miscarriage, foetal health, progesterone deficiency

## Abstract

**Background and Objectives:** Progesterone deficiency during pregnancy may lead to various complications, including first-trimester miscarriage, which is the most common pregnancy complication. However, progesterone therapy may play a role in pregnancy maintenance and foetal development. The aim of this scoping review is to present evidence on the link between progesterone deficiency and first-trimester miscarriage among pregnant women and assess the impact of progesterone therapy on foetal development. **Methods:** A comprehensive global systematic search of mainly primary research studies was conducted using several databases. Peer-reviewed studies published between 2010 and 2023 were included. The scoping review was conducted using the framework outlined by the Joanna Briggs Institute (JBI) and reported using the Preferred Reporting Items for Systematic Reviews and Meta-Analyses—Extension for Scoping Reviews (PRISMA-ScR) statement. **Results:** Twenty-three articles (which included 35,862 participants) were included in the analysis. Most studies were conducted in mid- to high-income countries. All 23 articles reported a significant positive relationship between progesterone deficiency and first-trimester miscarriage. Furthermore, the majority of studies reported a higher risk of miscarriage when lower levels of progesterone are combined with other declined hormones. While most studies reported that progesterone therapy may reduce the rate of first-trimester miscarriage among pregnant women, no evidence of health-related harm to offspring development was reported. **Conclusions:** The findings from this systematic–scoping review indicate possible benefits of progesterone replacement therapy in maintaining a healthy pregnancy and foetal development. Rigorous studies that include large sample sizes and systematic reviews are required to confirm these findings further.

## 1. Introduction

A threatened miscarriage (TM) refers to an ongoing pregnancy that is associated with vaginal bleeding, with or without abdominal pain. Symptoms can vary from blood spotting to potentially fatal shock. Once it proceeds to the dilation of the cervix, a miscarriage is inevitable [1,2]. Threatened miscarriage occurs in approximately 15–20% of all pregnancies and may lead to pregnancy complications including preterm delivery, foetal growth detainment, preeclampsia, eclampsia, the preterm premature rupture of membranes, placental abruption, and stillbirth in future pregnancies [3]. It is a predictor of long-term health issues such as cardiovascular disease and venous thromboembolism. Importantly, approximately 15% of those pregnancies result in a complete spontaneous miscarriage including first-trimester miscarriage [3,4]. First-trimester miscarriage is the most common pregnancy complication among pregnant women, and it tends to have a significant negative impact on a woman’s mental and physical health [1,5,6,7]. It has a significant negative impact on a woman’s physical health as it may cause bleeding and infection. It also has a significant negative impact on a woman’s psychological health as it may lead to anxiety, post-traumatic stress disorder, depression, and suicide [5,8].

Progesterone is an important hormone in early pregnancy. After the implantation stage into the uterine wall and the stimulation of human chorionic gonadotropin hormone (HCG), which is produced by the placenta, progesterone is secreted by the corpus luteum in the early stage of pregnancy [9,10].

Progesterone is responsible for oocyte maturation, embryo implantation, and supporting the placenta in early pregnancy. Therefore, insufficient progesterone levels in the luteal phase of menstruation and early gestation can lead to recurrent pregnancy loss [11,12]. Moreover, progesterone plays an essential role in regulating the maternal immune response, decreasing the uterine contraction, and enhancing utero-placental circulation. Therefore, in early pregnancy, the level of serum progesterone may be a predictor of pregnancy maintenance [13,14].

Wang et al. (2017) suggest that abnormal immune mechanisms are associated with the implantation of the allogeneic embryo. In particular, altered serum levels of interleukin 18 (IL-18), which is a proinflammatory cytokine that is synthesised by monocytes as well as macrophages, is associated with pregnancy complications such as failure of embryo implantation following in vitro fertilisation, recurrent miscarriage, preterm delivery, and pre-eclampsia. Therefore, changes to the IL-18 level may be associated with a growing risk of recurrent spontaneous miscarriage. Progesterone has been shown to decrease IL-18 secretion [15].

Additionally, progesterone plays an important role during implantation as it supports decidualisation, controls uterine contraction, and regulates maternal immune tolerance to the foetal semi-allograft [16]. Moreover, it stimulates lymphocytes to release progesterone-induced blocking factor (PIBF), which is an essential mediator that plays an important role in the control of anti-foetal immune responses throughout pregnancy [17,18]. Hence, progesterone deficiency is associated with TM [8].

In early pregnancy, the serum progesterone level may be a predictor of pregnancy maintenance [13,19,20]. Therefore, it is critical to determine the relationship between progesterone deficiency, especially in combination with other hormones, including oestrogen, HCG, PIBF, and the maternal immune system, such as the secretion of IL-18, and first-trimester miscarriage. In turn, it is important to assess the effectiveness of progesterone therapy for those who are at risk of miscarriage, including first-trimester miscarriage, and to assess the impact of progesterone therapy on pregnancy maintenance and foetal development. Identifying problems and managing them can prevent first-trimester miscarriage and further complications.

The aim of this scoping review is to study all available evidence to determine the association between progesterone deficiency and first-trimester miscarriage in women and to assess the impact of progesterone therapy on pregnancy maintenance and foetal development. This may allow the identification of knowledge gaps and the suggestion of further research directions.

## 2. Methodology

A comprehensive scoping review utilising the “Joanna Briggs Institute (JBI) methodology for scoping review” was performed, which is a search framework first suggested by Arksey and O’Malley in 2005 [21,22].

The scoping review consists of the following steps: developing the research question, identifying the relevant literature, selecting the relevant literature that meets the inclusion criteria, extracting results, and presenting these results [21].

A scoping review approach was chosen for this study, given the aim and objectives of the research. A scoping review design helps adopt a broader research strategy while also ensuring the reproducibility, transparency, and reliability of existing knowledge in the field. The review followed the methodology outlined by the Joanna Briggs Institute (https://jbi.global/scoping-review-network/resources# accessed on 2 March 2023) and adhered to the reporting guidelines provided by the Preferred Reporting Items for Systematic Reviews and Meta-Analyses—Extension for Scoping Reviews (PRISMA-ScR) statement [23]. The search strategy employed an iterative process and was guided by the following primary questions: ‘What is the association between progesterone deficiency and first-trimester miscarriage among pregnant women?’ and ‘what is the impact of progesterone therapy on pregnancy maintenance and foetal development?’.

A literature search was conducted In MEDLINE (PubMed), EBSCOhost, CINAHL, Cochrane, and ProQuest to identify papers published between 2010 and 2023, using a combination of keywords and MESH terms for progesterone deficiency and first-trimester miscarriage. Keywords included Progesterone AND (abortion, spontaneous OR miscarriage OR “pregnancy loss”) AND human. Other keywords included Progesterone therapy AND (abortion, spontaneous OR miscarriage OR “pregnancy loss” OR foetal development) AND human. Multiple databases were chosen for this study to improve results and reduce the risk of overlooking any eligible studies that could have been used during our final appraisal [24].

The titles and abstracts of studies were first screened against the above inclusion criteria to determine which articles would undergo full-text review. Then, the full text of the resulting papers was reviewed for inclusion. Furthermore, the reference list of all included articles was searched for additional articles. Articles that were considered for inclusion were those that included participants who were pregnant women at risk of first-trimester miscarriage (with or without vaginal bleeding) and had progesterone deficiency. Only peer-reviewed articles published in the English language between 2010 and 2023 were considered. Exclusion criteria included studies of women with multiple gestations and missed, incomplete, or inevitable miscarriages. We included articles that provided original data [e.g., randomised controlled trials and observational studies] as well as systematic reviews.

Once data extraction was completed, Braun and Clarke’s approach to thematic analysis was used to evaluate the data [25]. The approach consisted of six steps: 1. Being familiar with the data, 2. Producing initial codes for the data, 3. Searching for potential themes, 4. Reviewing themes, 5. Defining and naming themes, and 6. Reporting and analysing themes [25]. Phase 6 was completed using PRISMA-ScR guidelines [23].

Study data including study design, sample size, recruitment and setting, data collection method, and findings were extracted into a custom template developed in Microsoft Excel, and duplicates were excluded. Findings were further summarised through an iterative coding process and used to develop a series of categories that broadly captured the association between progesterone deficiency and first-trimester miscarriage and the impact of progesterone therapy on pregnancy maintenance and foetal development. Database searches; the screening of studies, including the selection process; and data extraction were conducted by M.B. and E.A.

## 3. Results

A total of 978 peer-reviewed articles were identified in the stated databases. However, following the removal of duplicates, only 415 articles underwent title and abstract screening, leading to 79 articles that underwent full-text evaluation. Of those articles, a total of 23 peer-reviewed papers (which included 35,862 participants) met all the criteria for inclusion in this review (see Figure 1).

### 3.1. Article Characteristics

One of the peer-reviewed publications was a double-blinded, placebo-controlled trial, while three publications were randomised controlled trials (RCTs). Furthermore, 11 of the analysed publications were prospective cohort and cross-sectional studies. The review also included two retrospective studies and six systematic reviews (see Table 1).

All included articles were published from 2010 onwards. There has been an overall increase in the number of studies published each year since 2017 (see Table 1). They all used quantitative methodology. Most studies were conducted in mid- to high-income countries, with only a few studies conducted in low-income countries. The top five countries where research was conducted were the United Kingdom (UK), Singapore, Germany, China, and Jordan. The remaining studies were spread across a number of countries, largely being one study per setting: China, Denmark, the Netherlands, Iran, Taiwan, Italy, Australia, the USA, India, Malaysia, Turkey, and South Korea. Recruitment and setting were conducted almost fully in hospitals and other healthcare settings.

A narrative account was prepared from the included studies to determine the association between progesterone deficiency and first-trimester miscarriage among pregnant women and to assess the impact of progesterone therapy on pregnancy maintenance and foetal development. The data were synthesised thematically into five themes. These are as follows:

### 3.2. The Association between Progesterone Level and Pregnancy Loss

Fourteen articles showed that there is an association between progesterone level and pregnancy loss [1,3,8,11,13,15,27,29,30,31,32,33,34,36]. Only one meta-analysis by Yan et al. suggested no significant association between progesterone deficiency and pregnancy loss [37].

### 3.3. Progesterone Level as a Predictor of Miscarriage

Seven studies showed a clear positive association between progesterone deficiency and first-trimester miscarriage. These articles collectively suggested that progesterone alone can be a sturdy predictor of early TM [8,11,15,30,31,32,33].

### 3.4. Progesterone Combined with Oestrogen or β-HCG or PIBF as Predictors of Miscarriage

Five studies showed that progesterone together with other hormones and factors including oestrogen, HCG, and PIBF may exacerbate the risk of first-trimester miscarriage [3,13,27,29,36].

### 3.5. Progestogen Therapy’s Effectiveness

Five articles showed the effectiveness of progesterone therapy in women who have TM, particularly in some subgroups such as women with one, two, or more previous miscarriages [1,26,28,34,35]. However, one meta-analysis and one placebo-controlled randomised clinical trial suggested that progesterone treatment is not effective in women who have TM [37,38].

### 3.6. The Impact of Progestogen Therapy on Foetal Development

A systematic review and meta-analysis of randomised, controlled trials (which included 388 randomised controlled trials) [39], a placebo-controlled randomised clinical trial [40], a prospective cohort study [41], and a systematic review [42] (which included 9 trials) showed that the use of progesterone therapy during pregnancy does not have a negative impact on foetal development. One systematic review and meta-analysis of randomised controlled trials showed that the use of progesterone in early pregnancy in those with history of recurrent miscarriage does not have a negative impact on foetal development. Rather, it was suggested that progesterone therapy may decrease the risk of low birth weight in later gestational weeks [39]. Also, a placebo-controlled randomised clinical trial [40], a prospective cohort study [41], and a systematic review [42] showed that progesterone therapy use in early pregnancy has no harmful (or beneficial) effects on foetal development.

## 4. Discussion

The aim of this scoping review was to present and summarise a comprehensive and descriptive analysis of the association between progesterone deficiency and first-trimester miscarriage among pregnant women and the impact of progesterone therapy on pregnancy maintenance and foetal development.

There have been many hypotheses that having a sufficient amount of progesterone may decrease the rate of pregnancy loss, including first-trimester miscarriage. The results of this scoping review demonstrate that there is an association between progesterone level and miscarriage rate. Fourteen articles showed clear evidence of an association between progesterone level and pregnancy loss [1,3,8,11,13,15,27,29,30,31,32,33,34,36]. Only one meta-analysis by Yan et al. suggested that there is no association between progesterone deficiency and pregnancy loss [37]. Further, seven studies showed that there is an association between progesterone deficiency and first-trimester miscarriage. These studies showed that progesterone alone can be a sturdy predictor of early TM [8,11,15,30,31,32,33]. Furthermore, five studies showed that progesterone together with other hormones and factors including oestrogen, HCG, and PIBF may be strong predictors of first-trimester miscarriage [3,13,27,29,36].

Furthermore, the results of this scoping review demonstrate that progesterone therapy use has a positive impact on pregnancy maintenance. Five articles showed the effectiveness of progesterone therapy in women who have TM, particularly in some subgroups such as women with one, two, or more previous miscarriages [1,26,28,34,35]. However, one meta-analysis and one placebo-controlled randomised clinical trial suggested that progesterone therapy is not effective in women who have TM [37,38]. Yan et al. suggested that there is no association between progesterone deficiency and pregnancy loss [37].

Also, the results of this scoping review demonstrate that progesterone therapy has no negative impact on foetal development. A systematic review and meta-analysis of randomised controlled trials, which included 388 randomised controlled trials [39], a placebo-controlled randomised clinical trial [40], a prospective cohort study [41], and a systematic review [42] (which included 9 trials) showed that the use of progesterone therapy during pregnancy does not have negative impact on foetal development. One systematic review and meta-analysis of randomised controlled trials showed that the use of progesterone in early pregnancy in those with history of recurrent miscarriage does not have a negative impact on foetal development. Rather, it was suggested that progesterone therapy may decrease the risk of low birth weight in later gestational weeks [39]. Also, a placebo-controlled randomised clinical trial [40] showed that progesterone therapy use in early pregnancy has no harmful effects on foetal development. The study further showed that there was no difference in the number of admissions to hospital or rates of diagnoses between children that were exposed to progesterone therapy during foetal development and those that were not exposed to progesterone therapy. Furthermore, a prospective cohort study [41] and a systematic review [42] showed that progesterone therapy use in early pregnancy has no harmful (or beneficial) effects on foetal development. This is promising given that evidence shows that progesterone therapy use has a positive impact on pregnancy maintenance.

Ultimately, the majority of studies that were included in this scoping review suggested that progesterone level is significantly associated with first-trimester miscarriage, or at least with the risk of first-trimester miscarriage, demonstrating that the level of this hormone can be used as a strong predictor of spontaneous or threatened miscarriage. Furthermore, these studies suggested that progesterone therapy may reduce the rate of first-trimester miscarriage among pregnant women without having any negative effect on foetal development. These findings concur with international guidelines which acknowledge the role of progesterone therapy in reducing the rate of first-trimester miscarriage among pregnant women [43].

## 5. Strengths and Limitations

To main strength of our scoping review is that we conducted a comprehensive literature search of good-quality peer-reviewed studies following gold-standard guidelines. However, there are some limitations. For our search strategy, we used a broad set of terms relating to first-trimester miscarriage and progesterone deficiency; therefore, it is possible that some studies were missed. Also, only studies in English were included. Therefore, studies published in other languages were not included in the summary of evidence. However, the authors do not imagine that substantial evidence has been overseen due to the above issues.

## 6. Conclusions and Recommendations

Overall, our scoping review illustrated that there is an association between progesterone deficiency and first-trimester miscarriage among pregnant women. There is consistent research evidence indicating a strong positive association between a decline in progesterone levels and first-trimester miscarriage among pregnant women. Our findings also suggest that the level of this hormone can be used as a strong predictor of spontaneous or threatened miscarriage. Furthermore, this scoping review illustrated that while progesterone therapy may reduce the rate of first-trimester miscarriage among pregnant women, it does not have any impact on foetal development.

Rigorous studies that include large sample sizes are needed in regard to the association between progesterone level and first-trimester miscarriage and the impact of progesterone therapy on pregnancy maintenance and foetal development. There needs to be more investigation on how progesterone relates to other factors such as the presence or absence of foetal heartbeat, the mother’s BMI, and some behavioural characteristics that impact miscarriage. A uniform core outcome set would aid future evidence synthesis.

## Figures and Tables

**Figure 1 children-11-00422-f001:**
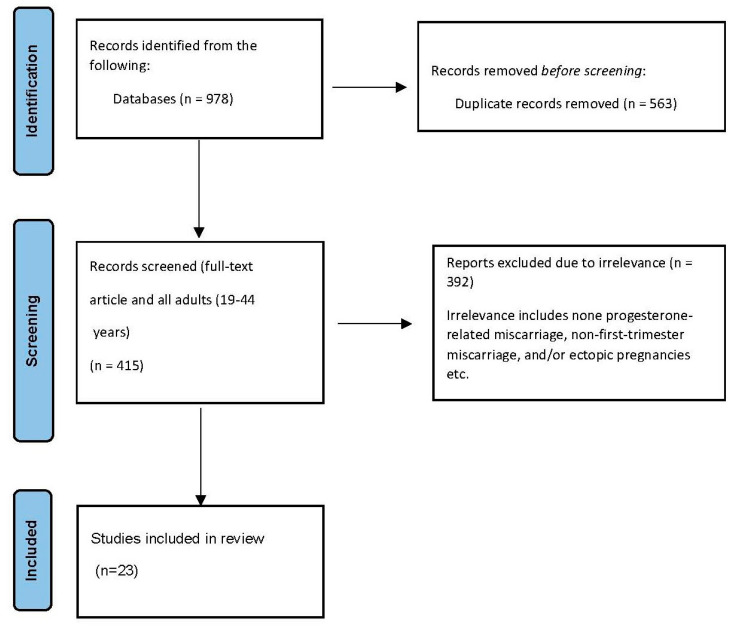
PRISMA flow chart of study selection.

**Table 1 children-11-00422-t001:** Characteristics of included studies.

Author, Year, and Country of Investigation	Study Title	Aim/Purpose	Population& Sample Size	Study Type	Results	Key Findings
Coomarasamy et al., 2020. UK. [26]	Progesterone to prevent miscarriage in women with early pregnancy bleeding: the PRISM RCT.	To assess the effects of vaginal micronised progesterone in women with vaginal bleeding in the first 12 weeks of pregnancy.	4153 participants.	A multicentre RCT	The live birth rate was 75% in the progesterone group and 72% in the placebo group.	Progesterone therapy given in the first trimester of pregnancy did not result in a significantly higher rate of live births among women with TM.
Deng et al., 2022. China. [27]	Prediction of miscarriage in first trimester by serum estradiol, progesterone and β-human chorionic gonadotropin (β-HCG) within 9 weeks of gestation.	To predict a miscarriage outcome within 12 weeks of gestational age by evaluating values of serum oestradiol, progesterone and β-HCG.	165 participants.	Retrospective study.	Progesterone levels at 7–9 weeks had an AUC of 0.766 (95% CI 0. 672–0.861), *p* = 0.000).	Low serum levels of oestradiol and progesterone or oestradiol alone at 7–9 weeks and β-HCG or combined progesterone and oestradiol at 5–6 weeks of gestation can be used better to predict first-trimester-miscarriage.
Devall et al., 2021. Australia, Germany, Hong Kong, the UK, and Singapore. [28]	Progestogens for preventing miscarriage: a network meta-analysis.	To estimate the relative effectiveness and safety profiles of the different progestogen treatments.	7 randomised trials (involving 5682 women).	A network meta-analysis.	Vaginal micronised progesterone may make little or no difference to the live birth rate when compared with a placebo in women with TM.	Progestogen may have a slight or no difference in threatened or recurrent miscarriage. However, vaginal micronised progesterone may increase the live birth rate.
Duan et al., 2010. China. [29]	Predictive power of progesterone combined with β-HCG measurements in the outcome of threatened miscarriage.	To investigate the predictive power of progesterone in combination with β-HCG measurements in the outcome of TM.	175 participants.	Retrospective study.	The mean serum levels of progesterone and β-HCG in patients with inevitable miscarriages were significantly lower than those in normal intrauterine pregnancies and ongoing pregnancies.	Progesterone combined with β-CG measurements may be useful for predicting the outcome of TM.
Haas et al., 2019. India, Jordan, the UK, and the USA. [1]	Progestogen for preventing miscarriage in women with recurrent miscarriage of unclear etiology.	To assess the efficacy and safety of progestogen as a preventative therapy against recurrent miscarriage.	Twelve trials (1856 participants).	A meta-analysis.	There may be a reduction in the number of miscarriages in women who are given progestogen supplementation compared to a placebo/controls.	For women with unexplained recurrent miscarriages, supplementation with progestogen therapy may reduce the rate of miscarriage in subsequent pregnancies.
Hanita and Hanisah, 2012. Malaysia. [30]	Potential use of single measurement of serum progesterone in detecting early pregnancy failure.	To determine the role of progesterone as a marker of early pregnancy failure.	95 participants.	A cross sectional study.	Progesterone levels were significantly lower in threatened abortion patients with outcomes of nonviable pregnancy compared with pregnancies that progressed on to the viability period.	Serum progesterone can be used as a marker for early pregnancy failure.
Kim et al., 2017. South Korea. [13]	Predictive value of serum progesterone level on β-HCG check day in women with previous repeated miscarriages after in vitro fertilization.	To assess the predictive value of the progesterone level on the β-HCG check day for ongoing pregnancy maintenance in in vitro fertilisation (IVF) cycles in women with previous unexplained repeated miscarriages.	148 participants.	Prospective observational study.	The overall ongoing pregnancy rate was 60.8%. The cut-off values of β-HCG levels higher than 126.5 mIU/mL and of progesterone levels higher than 25.2 ng/mL could be the predictive factors for ongoing pregnancy maintenance. The miscarriage rates were 19.5% (15/77) in the women with β-HCG > 126.5 mIU/mL and 13.0% (10/77) in those with >25.2 ng/mL.	The progesterone level at 14 days after oocyte retrieval can be a good predictive marker for ongoing pregnancy maintenance in women with repeated IVF failure with miscarriage, together with the β-HCG level. The combined cut-off value of progesterone > 25.2 ng/mL and β-HCG > 126.5 mIU/mL may suggest a good prognosis.
Ku et al., 2015. Singapore. [3]	How can we better predict the risk of spontaneous miscarriage among women experiencing threatened miscarriage?	To establish progesterone and PIBF levels as predictors of subsequent completed miscarriage among women presenting with TM between 6 and 10 weeks of gestation.	119 participants.	Prospective cohort study.	Low progesterone and PIBF levels are similarly predictive of subsequent completed miscarriage. Higher levels of progesterone and PIBF are linked to a lower risk of miscarriage, while low serum progesterone and PIBF levels predict spontaneous miscarriage among TM women between weeks 6 and 10.	Low progesterone and PIBF levels are predictive of subsequent completed miscarriage. Low serum progesterone and PIBF levels predict spontaneous miscarriage among women with TM between weeks 6 and 10.
Ku et al., 2018. Singapore. [8]	Serum progesterone distribution in normal pregnancies compared to pregnancies complicated by threatened miscarriage from 5 to 13 weeks gestation: a prospective cohort study.	To determine the distribution of maternal serum progesterone in normal pregnancies compared to those complicated by TM from 5 to 13 weeks’ gestation.	929 participants.	Prospective cohort study.	Median progesterone levels were lower in the TM cohort. In the subgroup analysis, the median serum progesterone concentration in women with ongoing pregnancy showed a linearly elevating trend from 5 to 13 weeks’ gestation. There was a non-significant elevation in serum progesterone from 5 to 13 weeks’ gestation in women who eventually had a spontaneous miscarriage.	Progesterone has a role in supporting early pregnancy, with lower serum progesterone being associated with TM and a subsequent complete miscarriage at 16 weeks’ gestation.
Ku et al., 2021. Singapore. [31]	Gestational age-specific normative values and determinants of serum progesterone through the first trimester of pregnancy.	To determine the gestational age-specific normative values of serum progesterone on a weekly basis, and its associated maternal and foetal factors, during the first trimester of a low-risk pregnancy.	590 participants.	A cross-sectional study.	There was an elevated level of serum progesterone during the first trimester and a transient decrease between gestational weeks 6 to 8. In women who had miscarried by 16 weeks, serum progesterone levels were significantly lower compared to women with viable pregnancies at 16 weeks for a mean difference of 24.2nmol/L.	Maternal age, BMI, parity, gestational age, and outcome of pregnancy at 16 weeks’ gestation may be linked to serum progesterone level.
Lek et al., 2017. Singapore. [32]	Validation of serum progesterone < 35 nmol/L as a predictor of miscarriage among women with threatened miscarriage.	To evaluate the validity of serum progesterone < 35 nmol/L with the outcome of spontaneous miscarriage by 16 weeks.	360 participants.	A prospective cohort study.	The study showed that the serum progesterone cut-off value of <35 nmol/L (11 ng/mL) is a useful predictor of miscarriage prior to week 16 of pregnancy.	As the serum progesterone cut-off value of <35 nmol/L (11 ng/mL) is a useful predictor of miscarriage prior to week 16 of pregnancy, patients can be quickly stratified as being at low or high risk of spontaneous miscarriage.
Tan et al., 2020. Singapore. [33]	Novel approach using serum progesterone as a triage to guide management of patients with threatened miscarriage: a prospective cohort study.	To assess the safety and efficacy of a clinical protocol using a certain serum progesterone level to prognosticate and guide the management of patients with TM.	1087 participants.	A prospective cohort study.	The miscarriage rate was 9.6% among 77.9% of participants with serum progesterone ≥ 35 nmol/L who were not treated with oral dydrogesterone. The miscarriage rate was 70.8% among women with serum progesterone < 35 nmol/L who were treated with dydrogesterone.	Patients with high serum progesterone levels can be reassured and counselled without medical treatment, while patients with low serum progesterone levels are at high risk of miscarriage even with treatment.
Turgal et al., 2016. Turkey. [34]	Effect of micronized progesterone on foetal-placental volume in first trimester threatened abortion.	To compare the effect of oral micronised progesterone on first-trimester foetal and placental volumes.	60 participants.	Randomised controlled trial.	After treatment, the placental volume difference was significantly higher in the oral micronised progesterone group (336%, 67–1077) than in the control group (141%, 29–900) (*p* = 0.007).	Hormonal support with oral micronised progesterone is associated with an elevated placental volume in first-trimester threatened abortion when compared to the control group.
Wahabi et al., 2018. Germany, Italy, Iran, Malaysia, Turkey, and Jordan. [35]	Progestogen for treating threatened miscarriages.	To determine the safety and efficacy of progestogens in the treatment of TM.	Seven trials (involving 696 participants).	Meta-analysis.	Treatment with oral progestogens decreases the miscarriage rate. Treatment with vaginal progesterone has little or no effect.	Progestogens may be effective in the treatment of TM. However, they may have little or no effect on the rate of preterm birth.
Wang et al., 2017. China. [15]	Effects of HCG, estradiol, and progesterone on IL-18 expression in human decidual tissues.	To evaluate the effects of oestradiol, HCG, and progesterone on IL-18 expression in human decidual tissues.	28 participants.	Prospective cohort study.	Oestradiol, HCG, and progesterone can reduce IL-18 secretion in the cultured endometrial stromal cells in patients with spontaneous abortion to the levels observed following normal pregnancy.	Progesterone can significantly decrease IL-18 expression and elevate the growth of CD56+ CD16− uNK cells. Therefore, it is suggested that these activities may underlie the mechanism by which progesterone enhances pregnancy outcomes.
Whittaker et al., 2018. The UK. [36]	Gestational hormone trajectories and early pregnancy failure: a reassessment.	To re-evaluate whether, in early pregnancy, gestational hormone trajectories can determine future miscarriage in asymptomatic pregnancies.	210 participants.	Prospective cohort study.	Progesterone and oestradiol displayed negative mean slopes in pregnancies destined for failure; in this group, both human placental lactogen (hPL) and HCG revealed mean positive trajectories that imitated normal pregnancies.	Oestradiol, progesterone, and HCG trajectories, from 50 days of gestation, have good potential for revealing pathophysiology and for identifying which asymptomatic pregnancies are destined for subsequent failure.
Yan et al., 2021. Taiwan. [37]	Efficacy of progesterone on threatened miscarriage: an updated meta-analysis of randomized trials.	To investigate the correlation between progesterone and improving pregnancy outcomes among those with TM.	4907 participants.	Meta-analysis.	Compared to the placebo, progesterone supplementation was associated with a decrease in the rate of miscarriage [RR = 0.70 95% Cl (0.52, 0.94)].	Progesterone supplementation did not significantly enhance the incidence of preterm and live birth.
Su et al., 2011. Taiwan. [11]	Association of progesterone receptor polymorphism with idiopathic recurrent pregnancy loss in the Taiwanese Han population.	To investigate the association between polymorphisms of the progesterone receptor gene and idiopathic recurrent pregnancy loss.	300 participants.	Prospective cohort study.	The allele and genotype frequencies of the functional SNP [PROGINS (rs1042838)] were both significantly higher in patients with idiopathic recurrent pregnancy loss than in the control subjects (both *p* = 0.006). In addition, the C-C haplotype, which consists of rs590688C>G and rs11224592T>C, is associated with a decreased risk of recurrent pregnancy loss (*p* = 0.004).	The PROGINS polymorphism confers susceptibility to idiopathic recurrent pregnancy loss in Taiwanese Han women.
Mclindon et al., 2023. Australia. [38]	Progesterone for women with threatened miscarriage (STOP trial): a placebo-controlled randomized clinical trial.	To determine the treatment effect of vaginal progesterone in women with TM.	556 participants.	A placebo-controlled randomised clinical trial.	The live birth rates were 82.4% and 84.2% in the intervention group and placebo group, respectively. Among women with at least one previous miscarriage, the live birth rates were 80.6% and 84.4%. Also, no significant effect was seen from progesterone in women with two or more previous miscarriages. The preterm birth rates were 12.9% and 9.3%, respectively.	No evidence to support the treatment effect of vaginal progesterone in women with TM.
Wu et al., 2021. China. [39]	Pregnancy-related complications and perinatal outcomes following progesterone supplementation before 20 weeks of pregnancy in spontaneously achieved singleton pregnancies: a systematic review and meta-analysis.	To assess the effects of progesterone supplementation before 20 weeks of pregnancy on pregnancy-related complications and perinatal outcomes in later gestational weeks.	Nine trials (involving 6439 participants).	Systematic review and meta-analysis of randomised, controlled trials.	The pooled odds ratio (OR) of low birth weight following oral dydrogesterone was 0.57 (95% CI 0.34–0.95, moderate quality of evidence).	The use of oral dydrogesterone in those with history of recurrent miscarriage, before 20 weeks of pregnancy, may decrease the risk of low birth weight in later gestational weeks.
Vedel et al., 2016. Denmark. [40]	Long-term effects of prenatal progesterone exposure: neurophysiological development and hospital admissions in twins up to 8 years of age.	To perform a neurophysiological follow-up in children exposed prenatally to progesterone at 48 or 60 months of age.	989 participants.	Placebo-controlled randomised clinical trial.	There was no difference in the number of admissions to hospital or rates of diagnoses between the groups. However, the ORs for a diagnosis concerning the heart were 1.66 (95% CI, 0.81–3.37), favouring the placebo.	Second- and third-trimester exposure of the foetus to progesterone may not cause long-term harmful effects.
Shen et al., 2015. China. [41]	Progesterone use in early pregnancy: a prospective birth cohort study in China.	To determine the effects of progesterone use in early pregnancy on birth and maternal outcomes.	6617 participants.	A prospective cohort study.	Women who used progesterone had significantly higher risks of caesarean section and post-partum depression. Progesterone use had no effect on preterm-birth prevention, foetal growth, or gestational diabetes.	Progesterone use in early pregnancy has no benefit or harm for specific pregnancy outcomes.
Simons et al., 2021. The Netherlands. [42]	The long-term effect of prenatal progesterone treatment on child development, behaviour and health: a systematic review.	To determine long-term effects in children after prenatal progesterone treatment.	388 papers, (involving 4222 participants).	Systematic review.	There was no difference in neurodevelopment as assessed by the Bayley-III Cognitive Composite score between those exposed to progesterone versus placebo. Other outcomes showed no differences.	There was no evidence of benefit or harm in offspring prenatally exposed to progesterone treatment for the prevention of preterm birth.

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
