# Peer review of "Exploring Progesterone Deficiency in First-Trimester Miscarriage and the Impact of Hormone Therapy on Foetal Development: A Scoping Review"

_children, 2024, doi:10.3390/children11040422_

Round 1

Reviewer 1 Report

Comments and Suggestions for Authors

Dear authors

congratulations the manuscript is of interest i have minor revisions to suggest

1) please check size and characters of words within the manuscript which looks different here and there

2) can you please add a reference to this sentence?

Threatened miscarriage occurs in approximately 15-20% of all pregnancies, and may lead to pregnancy complications including preterm delivery, foetal growth detainment, preeclampsia, preterm premature rupture of membranes, placental abruption, and stillbirth in future pregnancies.

3) please mentioning about preeclampsia remind also the most severe form and the ones that associate with worse maternal fetal outcome such as eclampsia (read and cite PMID: 35317697)

4)please check figure 1 is seems be shared by two pages

5) please sinthetize contents of table 1 to allow the reader to quickly understand the literature review evidence

6) please mention about international guidelines recommendation of progesteron for  Threatened miscarriage

best regards

Author Response

  1.  
  1. This has been amended accordingly.
  2. A reference has been added in page 1.
  3. Good point, thank you. This has now been added to page 1.
  4. This has been amended accordingly. Please see page 4.
  5. Thank you for the comment. We have now made table 1 more succinct to enable the reader to quickly understand the evidence. Furthermore, table 1 is summarised in the results and discussion sections.
  6. This has been added to page 12.

Reviewer 2 Report

Comments and Suggestions for Authors

Reviewer statement:

Exploring Progesterone deficiency and first-trimester miscarriage and whether hormone therapy impacts foetal development: A Scoping review

A threatened miscarriage (TM), defined as an ongoing pregnancy with vaginal bleeding and/or  abdominal pain and can lead to a miscarriage, occurring in approximately 15-20% of all pregnancies.  TM, in case of an ongoing pregnancy can lead to pregnancy complications. A first-trimester miscarriage is the most common pregnancy complication among pregnant women with a significant negative impact on a woman’s mental and physical health. Progesterone is an important hormone, responsible for oocyte maturation, embryo implantation, and supporting the placenta in early pregnancy. Insufficient progesterone levels in the luteal phase of menstruation and early gestation can lead to recurrent pregnancy loss.  The level of serum progesterone may be a predictor of pregnancy maintenance. The authors report a scoping review aiming to determine the association between progesterone deficiency and first-trimester miscarriage in women, and, to assess the impact of progesterone therapy on pregnancy maintenance and foetal development,  which is interesting, important and clinical relevant. This review can add valuable information to the readers, updating the available knowledge  and research on this topic.

Title: The title reflects the study reported and the type of study conducted, which is excellent.

Overall: The paper is written in clear and concise English making the article attractive to read. The authors should be complimented for doing this.

Abstract : see overall remarks and remarks throughout the article

Introduction:

The introduction section is attractive to read form a reader point of view, explaining the background and reason for conducting this study.

No comments on this section.

Methodology:

This section is well written and was easy to read. Despite , there are some points needing explanation and or clarification from a readers point of view to be able to understand and for the interpretation of the presented results.

1.      The authors report the performance of a scoping review. Form a readers  perspective, why not a “normal” systematic review? The added value of a scoping review instead of a systematic review should be thoroughly explained to the reader. Please do so.  

2.      The authors report the following om page 3: “ Articles that were considered for inclusion were: those that included participants who are pregnant women at risk of first-trimester miscarriage with or without vaginal bleeding, peer-reviewed articles, published in the English language, and published between 2010-2023. “  The authors should define pregnant women at risk of first-trimester miscarriage to the reader. When there is vaginal bleeding, it is obvious, but in women without bleeding, when were pregnant women considered at risk? Please elucidate thoroughly tom the reader as this is an important and crucial point.

3.      Furthermore, the relation with incomplete miscarriage should also be explained, as this is an important definition and exclusion criteria.

4.      Figer 1 is mentioned and discussed in this section, although, this is a result and should be mentioned in the result section. Please do so.

Results:

This section is well written and was easy to read. Despite , there are some points needing explanation and or clarification from a readers point of view to be able to understand and for the interpretation of the presented results.

5.      The result s are reported very brief and short. Elucidating more on the result is urgently suggested.

6.      The authors report the number of studies reporting an association. From a readers perspective, the number of studies is relevant, but more important are the number of participants included in those studies. Please report how many participants are compared in the conducted comparisons.

7.      Furthermore, the authors should conduct some statistics , to be able to define how strong the reported association is. Just the reporting of an association is not enough and a pity of all conducted work. Please report.

8.      The authors report the presence of a systematic review and meta-analysis including 388 randomized controlled trails. In comparison, the authors included 23 studies. Please elucidate on this enormous difference. Are the authors sure, that the search strategy they performed was well conducted? Please elucidate thoroughly on this point.

9.      With the knowledge of this systematic review, what was the aim of the authors, different than that of the systematic review?

Discussion:

This section is much too short. The authors do not elucidate enough on the presented work, the results and what this review adds to the current available knowledge. The clinical relevance is missing.

Tables and Figure

Figure 1: no comments

Table 1: an excellent table.

10.   The authors should add a second table in which table 1 is summarized with the most important result and table 1 is much too big for a reader to have a good overview.

Comments on the Quality of English Language

The language requires minor changes

Author Response

  1. We would like to firstly thank the reviewer for the positive remarks and feedback. 
  2. We appreciate that systematic reviews have more convincing evidence, however, a scoping review was preferred to help adopt a broader research strategy while also ensuring the reproducibility, transparency, and reliability of existing knowledge in the field. We have now added this section to the methodology for the choice of a scoping review design  
  3. This has been defined in page 3.
  4. Thank you for this important comment. With all due respect, we believe that this is out of the scope of this scoping review.
  5. This has been amended accordingly.
  6. Thank you for your comment. With all due respect, we believe that the extent to which the results were discussed is sufficient to answer the research question, following a scoping review structure.
  7. Thank you for your comment. We originally included the total number of participants in the abstract. However, now, the total number of participants has been also added in page 3.
  8. Thanks again for the important comment. We discussed this at the start of the work and most team members voted against including statistics as most scoping reviews do not focus on statistics but instead qualitative assessments of findings of different included studies. Statistics are usually used in higher level evidence such as meta-analysis and systematic reviews
  9. This issue has been rectified throughout the article (including in the abstract, page 3, and page 11) by replacing the word “studies” with “articles” where relevant, as some articles were systematic reviews that included a high number of studies.
  10. We apologise to the reviewer but we did not understand this comment/ question.
  11. Thank you for your comment. With all due respect, we believe that the extent to which the results were discussed is sufficient to answer our research question. Also, clinical relevance was discussed to the extent needed for our purpose, throughout the article. Please also see discussion section for further elaboration and interpretation of the findings of individual studies
  12. Thank you for your comment. We have now made table 1 more succinct to enable the reader to have a good overview.

Reviewer 3 Report

Comments and Suggestions for Authors

Dear authors, after reading your proposal I consider that it is well described and an extensive search of the literature was carried out, it is suggested that you improve the presentation of the table where the included studies are described, consider that they are presented horizontally, as well as be more specific in the search algorithm, and better describe the reasons for considering the 392 studies excluded due to irrelevance.

Author Response

we thank the review for the positive feedback and we would like to say that we have taken some of the other feedback on board to improve the manuscript including the reason for "irrelevance" for excluding the 392 studies. other improvements on the table/s have also been made. thank you

Round 2

Reviewer 2 Report

Comments and Suggestions for Authors

Exploring Progesterone deficiency and first-trimester miscarriage and whether hormone therapy impacts foetal development: A Scoping review

The authors have revised the article and addressed most of the raised points and comments provided by the reviewers. The authors should be complimented for this achievement. Despite, there are still important and crucial points needing explanation and / or clarification.

1.     On page 3, the authors report the following: “ Articles that were considered for inclusion were: those that included participants who are pregnant women at risk of first-trimester miscarriage with vaginal bleeding or without vaginal bleeding (for example, those at risk due to urinary tract infections, excessive alcohol or drug use, obesity, trauma, or injury), peer-reviewed articles, published in the English language, and published between 2010-2023  The definition of pregnant women at risk of first-trimester miscarriage remains puzzling. In case of vaginal bleeding, it is obvious, but is the risk for a miscarriage the same in women without bleeding, with urinary tract infections, excessive alcohol or drug use, obesity, trauma, or injury ? There is no reference, evidence given for this statement. Please provide evidence and discuss this in the discussion section to help the reader in the interpretation of  the provided risk factors.

2.     The authors defined the following question as out of the scope of this scoping review. Again, for the interpretation of the presented results, the relation with incomplete miscarriage should be explained thoroughly as this may influence the presented results. When was a miscarriage defined as incomplete and when were women than excluded if they were included at first, ( or were they not excluded). Important information to know and to discuss in the discussion section.

3.     The authors believe that the extent to which the results were discussed is sufficient to answer the research question, following a scoping review structure. The result are report very briefly. Despite a scoping review, the authors should at least try to report the result as scientific as possible. Only reporting that there is an association, with all due respect is must to little. As a reader, the magnitude of the associations should be reported, preferable with figures. The authors are urgently advice ta adapt the result section.

4.     Again, from a readers perspective and for interpretation of the presented results, the number of studies is relevant, but more important are the number of participants included in those studies. Please report how many participants are compared in the conducted comparisons.

5.     Furthermore, the authors should conduct some statistics , to be able to define how strong the reported association is. Just the reporting of an association is not enough and a pity of all conducted work. Please report despite that this was initially not the aim.

6.     The authors answer the following on the question:  The authors report the presence of a systematic review and meta-analysis including 388 randomized controlled trails. In comparison, the authors included 23 studies : This issue has been rectified throughout the article (including in the abstract, page 3, and page 11) by replacing the word “studies” with “articles” where relevant, as some articles were systematic reviews that included a high number of studies. The essence remains, how could the difference between the systematic review and the current study be explained. A crucial point.

AAs a reviewer, may I urge the authors to reply on the raised point to improve the quality and increase the scientific quality of the article, thereby facilitating the reader.

Author Response

The authors thank the reviewer for their time and efforts and persistence to further improve the manuscript. we hope they are now satisfied with our responses and adjustments and that they will also accept our views as authors who technically ticked all boxes to make sure this scoping review abides by the standard framework for scoping reviews and provides a broad scope of the issue at hand.  

Please see a new point-by-point response to the comments below:

  1. On page 3, the authors report the following: “ Articles that were considered for inclusion were: those that included participants who are pregnant women at risk of first-trimester miscarriage with vaginal bleeding or without vaginal bleeding (for example, those at risk due to urinary tract infections, excessive alcohol or drug use, obesity, trauma, or injury), peer-reviewed articles, published in the English language, and published between 2010-2023”  The definition of pregnant women at risk of first-trimester miscarriage remains puzzling. In case of vaginal bleeding, it is obvious, but is the risk for a miscarriage the same in women without bleeding, with urinary tract infections, excessive alcohol or drug use, obesity, trauma, or injury ? There is no reference, evidence given for this statement. Please provide evidence and discuss this in the discussion section to help the reader in the interpretation of  the provided risk factors.

Response: we thank the reviewer for raising this point which seems quite a fair point in the context of miscarriage and based on the wording of the section on inclusion criteria in this paper. however, unfortunately, it seems that we have included this as “inclusion criteria” by error. Our inclusion criteria were strictly based on our keywords in the previous paragraph of the methodology section where the word “progesterone” AND any “1st-trimester miscarriage” was a must in any study to be selected as stated below: “A literature search was conducted in MEDLINE (PubMed), EBSCOhost, CINAHL, Cochrane, and ProQuest to identify papers published between 2010-2023, using a combination of keywords and MESH terms for progesterone deficiency and first-trimester miscarriage. Keywords included Progesterone AND (abortion, spontaneous OR miscarriage OR “pregnancy loss”) AND human. Other keywords included Progesterone therapy AND (abortion, spontaneous OR miscarriage OR “pregnancy loss” OR foetal development) AND human.”  Since the review focuses only on 1st-trimester miscarriage due to “progesterone deficiency”.  Please note that adding reasons for miscarriage including “vaginal bleeding” or “no vaginal bleeding” has never been essential to our search criteria and this confusing statement has now been omitted from the “inclusion criteria”. The statement in the section now reads as follows “Articles that were considered for inclusion were those that included participants who are pregnant women at risk of first-trimester miscarriage (with or without vaginal bleeding) and had progesterone deficiency.”

  1. The authors defined the following question as out of the scope of this scoping review. Again, for the interpretation of the presented results, the relation with incomplete miscarriage should be explained thoroughly as this may influence the presented results. When was a miscarriage defined as incomplete and when were women than excluded if they were included at first, ( or were they not excluded). Important information to know and to discuss in the discussion section.

This is a good point. However, if we understand the point correctly, some studies included in this review assessed the level of progesterone in women who had a “first-trimester miscarriage” or had “threatened first-trimester miscarriage” i.e. regardless of whether miscarriage was complete or incomplete, compared to those who had normal pregnancies.  We have now added a line in the discussion to further highlight this issue “Ultimately, the majority of studies that were included in this scoping review suggested that progesterone level is significantly associated with first-trimester miscarriage, or at least with the risk of first-trimester miscarriage, demonstrating that the level of this hormone can be used as a strong predictor of spontaneous or threatened miscarriage”.

  1. The authors believe that the extent to which the results were discussed is sufficient to answer the research question, following a scoping review structure. The result are report very briefly. Despite a scoping review, the authors should at least try to report the result as scientific as possible. Only reporting that there is an association, with all due respect is must to little. As a reader, the magnitude of the associations should be reported, preferable with figures. The authors are urgently advice ta adapt the result section.

Again, good suggestion, However, we believe this is more suited for a systematic review and meta-analysis, for example the magnitude of the association, etc. we have recently published a scoping review in IJERPH (MDPI Q1 journal) where one of the requests made by the “editorial team members” was that the results section should be succinct and straight to the point and with no extra information as this will be done in the discussion section. We subsequently trimmed the results section further as it is designed to only include a summary of findings related to the research question. Our research question was the impact of progesterone deficiency on 1st-trimester miscarriage and the impact of progesterone therapy on the health of the foetus and child. This is a descriptive analysis so we hope we will be spared any statistical work as we summarise the evidence in a more qualitative manner. This qualitative analysis typically involves identifying common themes and patterns in the literature to provide an overview of the existing evidence and highlight areas for further research. Therefore, scoping reviews tend to prioritize qualitative synthesis over statistical analysis. https://bmcmedresmethodol.biomedcentral.com/articles/10.1186/s12874-018-0611-x

  1. Again, from a readers perspective and for interpretation of the presented results, the number of studies is relevant, but more important are the number of participants included in those studies. Please report how many participants are compared in the conducted comparisons.

We have addressed this important comment and added the total number of all participants in the analysed studies. Please see page 3 last paragraph (results section)- this is highlighted in yellow in the manuscript.  

  1. Furthermore, the authors should conduct some statistics , to be able to define how strong the reported association is. Just the reporting of an association is not enough and a pity of all conducted work. Please report despite that this was initially not the aim.

We are sincerely sorry but the scoping review we have developed does not include statistical analysis and only qualitative thematic analysis aiming at identifying common themes and patterns. It is recommended that scoping review analysis sticks to descriptive analysis rather than statistical. See the evidence below in relation to this and other aspects of Scoping review methodologies and principles. Please also see point no. 3 above

Peters MDJ, Godfrey C, McInerney P, Munn Z, Tricco AC, Khalil, H. Chapter 11: Scoping Reviews (2020 version). In: Aromataris E, Munn Z (Editors). , JBI, 2020. Available JBI Manual for Evidence Synthesis from . https://synthesismanual.jbi.global https://doi.org/10.46658/JBIMES-20-12

Also, see the following in relation to comparing systematic reviews and scoping reviews:

https://med.cornell.libguides.com/systematicreviews/scopingreviews#:~:text=Scoping%20reviews%20aim%20to%20provide,%2Fstatistical%20pooling%20is%20performed

https://bmcmedresmethodol.biomedcentral.com/articles/10.1186/s12874-018-0611-x

based on the above, we hope the reviewer will understand our point of view. Conducting statistical analysis is a major task and one we have not done in any of our scoping reviews in the past.

  1. The authors answer the following on the question:  The authors report the presence of a systematic review and meta-analysis including 388 randomized controlled trails. In comparison, the authors included 23 studies : This issue has been rectified throughout the article (including in the abstract, page 3, and page 11) by replacing the word “studies” with “articles” where relevant, as some articles were systematic reviews that included a high number of studies. The essence remains, how could the difference between the systematic review and the current study be explained. A crucial point.

Another good question. Some scoping reviews can be focused on primary research studies while others can include both primary and secondary, as they tend tobe less “systematic” than say a systematic review, which we think provides more broad scope and summary of evidence. Our scoping review of this important clinical issue has combined evidence from primary analysis (observational and experimental studies) and secondary analysis (systematic reviews and meta-analysis). We think this adds strength to the evidence although raises questions in relation to the heterogeneity of studies, which is not necessarily an issue for scoping review analysis.

AAs a reviewer, may I urge the authors to reply on the raised point to improve the quality and increase the scientific quality of the article, thereby facilitating the reader.

Thank you, we have now done this to the best of our knowledge and as per scoping reviews protocol.